# Monoclonal antibody-based immunohistochemistry reveals residual *Taenia solium* antigens in calcified granulomas from pigs with neurocysticercosis

Luz M. Toribio[1,2]*, Lizziee B. Tello-Ccente[2], Gianfranco Arroyo[2,3]*, Manuela R. Verastegui[2], Robert H. Gilman[4], Theodore E. Nash[5], Hector H. Garcia[2,4,6], Javier A. Bustos[2,6] on behalf the Cysticercosis Working Group in Peru [CWGP]¶

1 Institute for Infection and Immunity, School of Health and Medical Sciences, City St George's, University of London, London, United Kingdom, 2 Center for Global Health, Parasitological Diagnostic Laboratory, Laboratories of Research and Development, Faculty of Sciences and Philosophy, and Infectious Diseases Research Laboratory, Laboratories of Research and Development, Faculty of Sciences and Philosophy, Universidad Peruana Cayetano Heredia, Lima, Peru, 3 Facultad de Medicina Veterinaria y Zootecnia, Universidad Cientifica del Sur, Lima, Peru, 4 Department of International Health, Bloomberg School for Public Health, Johns Hopkins University, Baltimore, Maryland, United States of America, 5 Laboratory of Parasitic Diseases, National Institute of Allergy and Infectious Diseases, National Institutes of Health, Bethesda, Maryland, United States of America, 6 Cysticercosis Unit, Instituto Nacional de Ciencias Neurologicas, Lima, Peru

¶ Membership of the Cysticercosis Working Group in Peru [CWGP] is listed in the Acknowledgments.
* gianfranco.arroyo@upch.pe (GA); ltoribio@sgul.ac.uk (LMT)

## Abstract

### Background

Neurocysticercosis (NCC), a parasitic brain infection caused by *Taenia solium* larvae, remains a leading cause of preventable epilepsy globally. Although calcified brain lesions were formerly considered as the quiescent end stage of NCC, they may act as epileptogenic foci. It has been suggested that parasitic antigens within calcified lesions may act as potential triggers of inflammation and subsequent seizure activity. In this study, we developed and optimized immunohistochemistry (IHC) assays employing anti-*Taenia solium* monoclonal antibodies (mAbs) to detect residual cyst antigens in calcified lesions in a porcine NCC model and assessed antigen persistence for up to 12 months after successful antiparasitic treatment.

### Methods/principal findings

Six mAbs raised against *T. solium* whole cyst (TsW5, TsW8, and TsW12), vesicular fluid (TsV3 and TsV4), and excretory/secretory products (TsE1) were used for IHC assay development and tested in brain sections containing viable brain cysts from NCC pigs and uninfected tissue from controls to optimize assay conditions, blocking, primary and secondary antibody dilutions. Optimized assays were subsequently performed in selected calcified granulomas (*n* = 20) obtained from NCC-infected pigs

**Data availability statement:** All data are available without restriction in the Supporting Information Files.

**Funding:** This study partially was funded by the National Institute of Allergy and Infectious Diseases (NIAID)-National Institutes of Health (NIH, grant number R01AI 116456, to JAB), and The Fogarty International Center (FIC)-NIH (grant number D43TW001140, to HHG). The funders had no role in the study design, data collection and analysis, decision to publish or preparation of the manuscript.

**Competing interests:** The authors have declared that no competing interests exist.

sacrificed at 4, 8, and 12 months after antiparasitic treatment to identify residual cyst antigens as well as their localization and area of reactivity. We observed residual cyst antigens in 65–80% of calcified granulomas, with TsW8 and TsV3 showing the highest percentages of immunoreactivity. Antigen localization followed two patterns, one with antigens entirely located within the calcified lesions (TsW5, TsW8, TsW12, and Tsv4) and another with antigens located outside the cyst in the perilesional brain tissue (TsV3 and TsE1). Antigen detection and the extent of reactivity declined progressively after antiparasitic treatment but persisted at detectable immunoreactive areas in calcified granulomas up to month 12 months after treatment.

## Conclusions/significance

*T. solium* antigens remain detectable in calcified granulomas and in the perilesional tissue for up to 12 months after antiparasitic treatment in the pig model.

## Author summary

Neurocysticercosis (NCC) is a parasitic infection of the brain caused by the larval stage of *Taenia solium* and a leading cause of preventable epilepsy worldwide. Following parasite death, lesions frequently calcify in the brain. Calcified NCC lesions were long considered inactive lesions. However, growing evidence indicates that calcifications may still contribute to seizure development. One proposed mechanism involves the persistence of parasite antigens trapped in calcified lesions, which may intermittently trigger inflammation in the adjacent brain tissue. In this study, we developed and optimized immunohistochemical assays using specific anti-*T. solium* monoclonal antibodies to detect residual cyst antigens in calcified brain lesions from pigs with NCC, a well-established animal model of the human disease, evaluated at three time points after antiparasitic treatment. IHC demonstrated persistent parasite antigens in most calcified lesions for up to 12 months post-treatment, and occasional detection in surrounding tissue. Although antigens declined over time, they did not fully resolve.

## Introduction

Neurocysticercosis (NCC) is a parasitic infection of the central nervous system (CNS) caused by the larval stage (cysticercus) of *Taenia solium*. It represents one of the most common causes of adult-onset epilepsy globally and a significant public health concern [1,2]. NCC is prevalent in many low-and middle-income countries, where it accounts for up to 30% of adult epilepsy cases [3–5]. Moreover, imported NCC cases are frequently diagnosed in high-income countries because of migration from endemic areas, adding a significant burden on healthcare systems [6–9]

The lifecycle of *T. solium* involves humans as definitive hosts, carrying the adult tapeworm (taeniasis), and pigs as intermediate hosts, harbouring cysticerci in the musculature and other organs after ingestion of eggs released in the faeces from a tapeworm carrier. Humans acquire cysticercosis by accidental ingestion of *T. solium* eggs by the faecal-oral route from close contact with a tapeworm carrier, or maybe by other mechanisms of infection including contaminated food, water, fomites, and less likely, self–autoinoculation). When cysts establish in the CNS, they produce NCC, and patients can remain asymptomatic for many years before neurological symptoms appear, most commonly seizures [10,11].

Initially, parenchymal viable brain cysts produce very little or no inflammation through parasite-mediated immunomodulation mechanisms [12]. However, when cysts start to degenerate (either spontaneously through natural evolution or induced by antiparasitic treatment), a pronounced inflammatory cascade characterized by reactive astrocytosis, microglial activation, and disruption of the blood-brain barrier occurs [13]. This process may lead to cyst clearance or to the development of residual calcifications in around 40% [14]. Formerly perceived as inactive lesions, residual NCC calcifications can show enhancement and perilesional edema on neuroimaging (MR) in patients at the time of seizure episodes [13–15], although a 29.2% of perilesional edema episodes can be by asymptomatic and detected on MRI during routine follow–up [15].

The exact mechanisms by which calcified NCC lesions could cause seizures remain incompletely understood, but current evidence suggest they are multifactorial and involve both immunological and structural brain changes [16,17]. Even long after parasite death, transient perilesional inflammatory episodes may occur around calcified granulomas, while the chronic presence of these calcified granulomas can induce gliotic remodelling of the surrounding brain cortex, promoting the formation of hyperexcitable neuronal circuits with increased excitability that facilitates epileptogenesis. A proposed mechanism also suggests that inflammatory episodes are triggered by the presence of residual cyst antigens trapped in calcified lesions, which can intermittently stimulate the host's immune response [18,19]. Histopathological studies in NCC have revealed tissue remnants of scolex and hooks in calcified granulomas, indicating that cyst components can remain trapped inside lesions [11,20–24]. Nonetheless, progress in clarifying this process is hindered by the limited availability of human brain biopsies and the absence of reliable assays for detecting *T. solium* tissue antigens.

We developed and optimized immunohistochemistry (IHC) assays using a series of in-house anti-*T. solium* mAbs [25] to identify residual cyst antigens in calcified granulomas from our archive records of naturally infected pigs with NCC [26]. Our findings provide novel insights into the calcification process in NCC, emphasizing cyst antigen presence and persistence in calcified lesions and surrounding cerebral tissues.

## Materials and methods

### Ethics statement

This study was approved by the Institutional Ethics Committee for Animal Use and Care at Universidad Peruana Cayetano Heredia (approval number: R–036–12–18). All tissue samples used in this study were obtained from our archive biological bank from previous experiments conducted in accordance with the standard guidelines of the Association for Assessment and Accreditation of Laboratory Animal Care (AAALC/NIH).

### Antibody selection

From 21 in-house anti-*T. solium* mAbs [25] we selected six mAbs targeting whole cyst (TsW5, TsW8, and TsW12), vesicular fluid (TsV3, and TsV4), and excretory/secretory products (TsE1). These mAbs were selected according to their specific recognition patterns observed in histological sections and strong reactivity in both serum and urine as previously reported [25]. These mAbs were isotypes IgM (TsW5, Tsw8, and TsE1), IgG1 (TsV3, and TsV4), or IgG3 (TsW12) and culture supernatants (RPMI medium) of mAb-producing hybridomas were screened by direct ELISA for detection of total *T. solium* antigens [25] and used directly as primary antibodies in our IHC assays.

## Tissue samples

We used a total of 32 paraffin-embedded brain biopsies of pigs from our archive records. For the development and optimization of IHC assays, six biopsies containing viable brain cysts from two untreated naturally infected NCC pigs were selected as positive tissue controls, whereas six brain biopsies from two uninfected pigs served as negative controls for pericystic brain tissue. To evaluate the presence of residual *T. solium* antigens in calcified lesions, we used 20 brain biopsies from a cohort of eight NCC-infected pigs treated with albendazole (15 mg/kg for 5 days) plus praziquantel (75 mg/kg for 1 day) and sacrificed at 4, 8, and, 12 months post-treatment (*n* = 4, 10, and 6 brain biopsies respectively) [26]. Biopsies contained calcifications were first screened on computed tomography (CT) imaging and subsequently confirmed as calcified granulomas by histopathological examination using Hematoxylin-Eosin and Alizarin Red staining for calcium identification. Sample selection was based on the availability of archived specimens that met the following criteria: biopsies containing well-preserved parasite material, while excluding paraffin-embedded biopsies with cracks or air bubbles.

## Study activities

**Tissue sectioning and slide preparation.** Paraffin–embedded biopsy specimens were serially sectioned on a microtome to a uniform thickness of 4 µm. From each block, central sections (defined as those containing well–delineated cystic structures and areas of calcified granulomas) were selected and mounted onto poly-L-lysine–coated slides (Thermo Scientific, Waltman, MA, USA).

**Deparaffinization and rehydration of tissue sections.** Following mounting, slides were placed in a pre–heated oven at 65ºC for 15 minutes to facilitate wax softening and initial tissue adherence. Then, slides underwent a standard xylene treatment for 5 minutes to remove paraffin wax. Immediately after xylene treatment, tissue slides were transferred back into the oven and incubated at 65ºC for 15 minutes to ensure complete evaporation of residual xylene. Subsequently, a graded ethanol rehydration series was applied, consisting of sequential immersions in 100%, 96%, 70% ethanol (each for 5 minutes), followed by a final rinse in distilled water for 5 minutes. Deparaffined slides were stored at 4ºC for no longer than 24 hours prior to IHC assay.

**Development and optimization of IHC assays.** Tissue slides underwent antigen retrieval in citrate-Tween 20 buffer (0.05%, pH 6) at 95ºC for 8 minutes, followed by inhibition of endogenous peroxidase activity with PBS 1X + 30% $H_2O_2$ for 30 minutes. For optimization, titration of blocking buffers, primary antibodies, and secondary antibodies was performed (see S1 Table). Non-specific binding was reduced by adding a blocking solution containing PBS-Triton X-100 (0.1%), Tween-20 (0.05%), skimmed milk (6%) and goat serum (5% or 10%), followed by three washing steps with PBS 1X-Tween 20 (0.05%). Between 300–700 µl of primary antibodies (anti-*T. solium* mAbs) at varying dilutions were added to the slides to cover the entire tissue area and incubated overnight (16–24 hours) in a humid chamber at 4ºC. After completion of eight washing steps, slides were incubated with 300–700 µL of biotin–labelled goat anti–mouse IgG/IgM secondary antibody (Thermo Scientific, Waltham, MA, USA) for 1 hour at room temperature. To assess non–specific binding of the secondary detection system, an additional slide was processed in parallel from which the primary antibody had been omitted. Streptavidin solution 1/400 was prepared and subsequently added, incubated for 30 minutes, and washed two times. Finally, a 3'3 diaminobenzidine (DAB) solution 1:100 was added for 2 minutes for colour development, after which the reaction was stopped with PBS-1X. Haematoxylin solution (Sigma-Aldrich, USA) was applied as counterstain for 30 seconds. Slides were subsequently dehydrated in graded alcohol series and mounted with Entellan medium for microscopic evaluation. Qualitative assessment of IHC assay optimization was performed by visual inspection, considering increased immunorreactivity and reduced background in viable cysts, together with the absence of immunorreactivity in control tissue. The optimized IHC assays were then used to evaluate the presence of residual cyst antigens in calcified granulomas.

   **Imaging processing.** All slides were examined using a light microscope (Primo Star, Zeiss) coupled to a 5.3 MP Basler camera (Microvisioner) to evaluate parasite antigens recognized by each mAb in the IHC assays. Antigen recognition patterns for each mAb in calcified granulomas were also described. Microphotographs at 20X magnification were taken, and a region of interest (ROI) encompassing the entire calcified granuloma was delineated, along with an additional ROI covering the surrounding brain tissue (~500 μm). The areas of immunoreactivity for each mAb-based IHC assay were quantified using the FIJI Open Access Image Processing Program (ImageJ, Maryland, USA), through image deconvolution, followed by conversion to an 8-bit white/black scale (0–255 pixels) with a threshold of 120. The percentages of areas showing immunoreactivity to *T. solium* antigens were calculated relative to the entire granuloma area and the perilesional tissue area. Sample processing and image analyses were performed under blinded conditions regarding timepoints after antiparasitic treatment.

## Statistical analysis

The overall proportion of calcified granulomas exhibiting immunoreactivity to *T. solium* antigens and their distribution by post-treatment timepoints (4, 8, and 12 months) were described and compared using Fisher's Exact test. The immuno-reactive areas to *T. solium* antigens for each mAb were summarized using mean ± standard errors (SE), distributed by post-treatment timepoints and analyzed using the non-parametric Cuzick test for trend. Statistical analyses were carried out in Stata SE v18.0 (Stata Corp., College Station, TX, USA) and plots were generated in GraphPad Prism v9.5.1 (GraphPad Software, LLC). A $P$ value < 0.05 was considered statistically significant.

## Results

### IHC assay development and optimization

Concentrations of blocking agents were adjusted according to the degree of the background staining in negative controls (S1 Table). The optimal dilution of primary antibodies (mAbs supernatants) varied considerably, ranging from 1:5 for TsW12 to 1:200 for TsW12, while secondary antibodies showed optimal performance at dilutions between 1:500–1:700. All tissue sections containing viable brain cysts demonstrated strong immunoreactivity to parasite antigens in the IHC assays, confirming the assay sensitivity and robustness of the protocols (Fig 1A–1F). None of the negative tissue controls and controls of secondary antibody detection system showed nonspecific staining, further demonstrating the reliability of assay conditions (Fig 1G –1I)

### Evaluation of residual parasite antigens in calcified granulomas

Overall, IHC assays showed high percentages of immunoreactivity to *T. solium* antigens in calcified granulomas. The proportion of positive biopsies ranged from 65–80%, with TsW8 and TsV3 mAbs showing the highest percentages of reactivity (80%), followed by TsW12 (75%), TsW5 (70%), and TsV4/TsE1 (64%, Table 1). At four months post-treatment, all evaluated biopsies were positive to *T. solium* antigens for several markers (TsW5, TsW8, TsW12, TsV4, and TsE1), while percentages of reactivity tended to decrease gradually at later post-treatment timepoints (8 and 12 months). However, none of these differences reached statistical significance.

   Two distinct patterns of antigen recognition were identified in calcified granulomas using our IHC assays. The mAbs TsW5, TsW8, TsW12, and TsV4 recognized *T. solium* antigens within the calcified granulomas (Fig 2A–2C). In contrast, a second pattern was observed with the mAbs TsV3 and TsE1, which recognized *T. solium* antigens distributed in the brain tissue surrounding the granuloma (Fig 2D–2F).

   The largest immunoreactive areas to *T. solium* antigens within calcified granulomas were detected using mAbs TsW8 and TsW12, whereas TsV3 and TsE1 showed the lowest reactivity (S2 Table). Immunoreactive areas within calcified granulomas exhibited a progressive decline across post-treatment time points for all mAbs, except TsV3 (Fig 3A), although

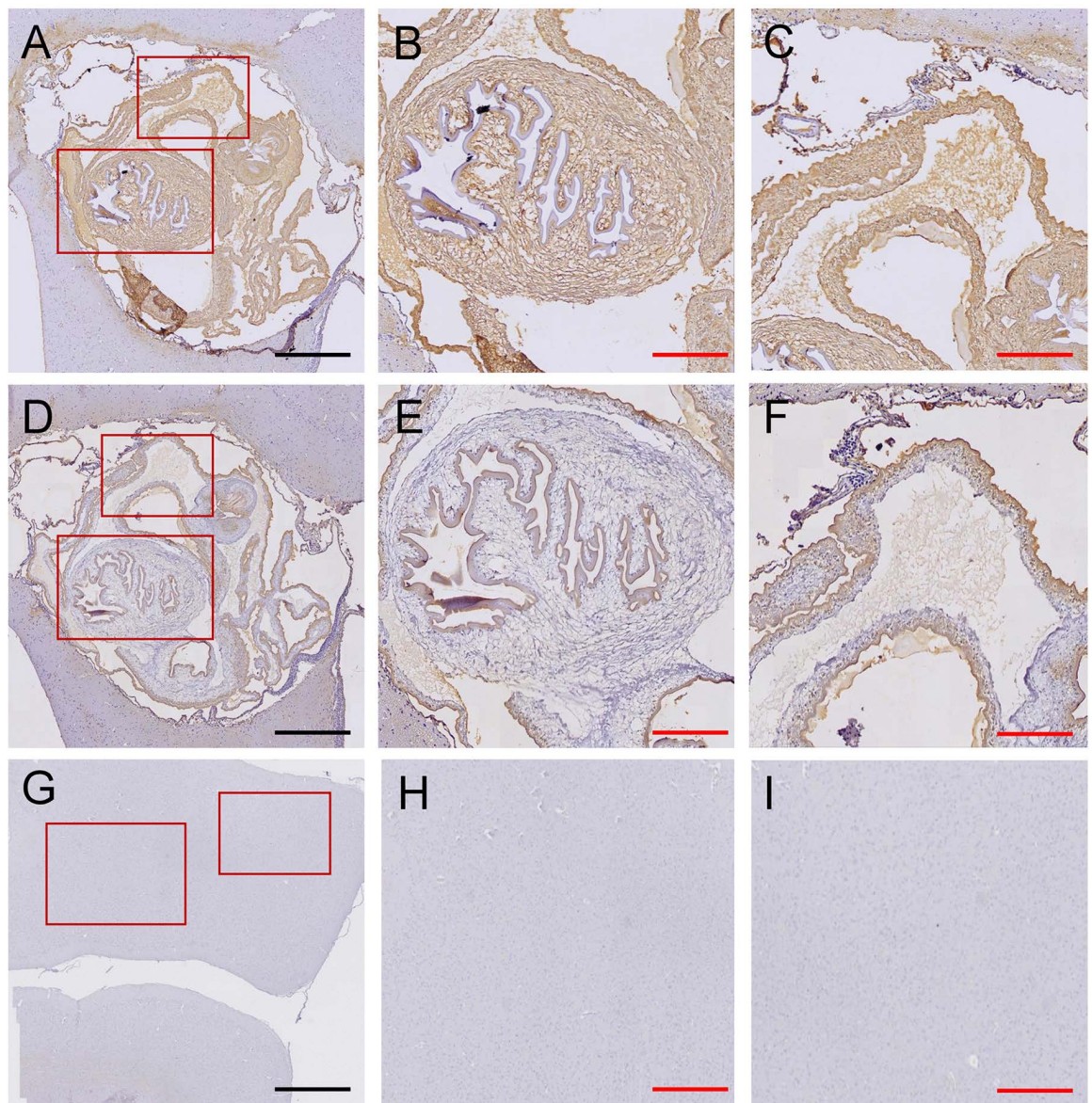

**Fig 1. Antigen recognition of mAb-IHC assays in viable brain cysts using mAbs TsW8 (A–C) and TsV3(D–F); lack of antigen recognition in brain cortex from an uninfected pig (G–I).** Scale black bar (400 µm); scale red bar (200 µm).

the trend analyses did not achieve statistical significance. In contrast, the brain tissue surrounding calcified granulomas showed larger immunoreactive areas with mAb TsV3 and TsE1. Trend analyses revealed a statistically significant decreasing pattern in perilesional immunoreactive areas according to post-treatment time points for both TsV3 (P for trend = 0.010) and TsE1 (P for trend = 0.022), with low but still detectable immunoreactive areas observed in biopsies up to 12 months post-treatment (S2 Table and Fig 3B).

**Table 1. Percentages of calcified granulomas from naturally infected NCC pigs showing positive immunoreactivity to residual cyst antigens detected by mAb-based IHC assays, and distribution across post-treatment timepoints.**

| *T. solium* mAbs | Total (*n*=20) | Post-treatment time points | | | *P** |
| --- | --- | --- | --- | --- | --- |
| | | 4 months (*n*=4) | 8 months (*n*=10) | 12 months (*n*=6) | |
| TsW5 | 14 (70.0) | 4 (100.0) | 7 (70.0) | 3 (50.0) | 0.294 |
| TsW8 | 16 (80.0) | 4 (100.0) | 8 (80.0) | 4 (66.7) | 0.628 |
| TsW12 | 15 (75.0) | 4 (100.0) | 7 (70.0) | 4 (66.7) | 0.640 |
| TsV3 | 16 (80.0) | 4 (100.0) | 8 (80.0) | 4 (66.7) | 0.628 |
| TsV4 | 13 (65.0) | 3 (75.0) | 7 (70.0) | 3 (50.0) | 0.698 |
| TsE1 | 13 (65.0) | 4 (100.0) | 7 (70.0) | 2 (33.3) | 0.119 |

**P* values obtained from Fisher's Exact test

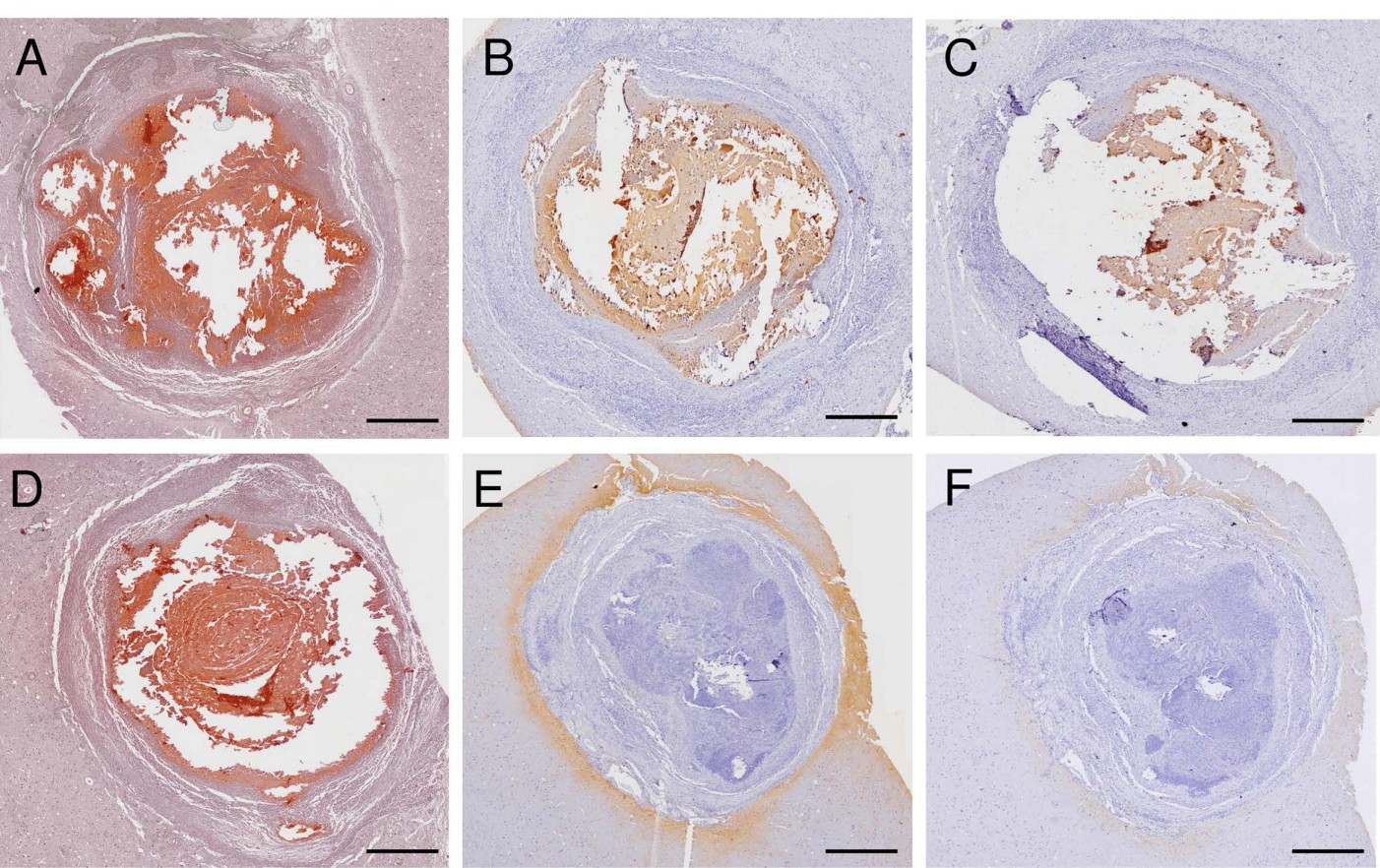

**Fig 2. Immunohistochemical analysis of neurocysticercotic calcified granulomas revealed distinct spatial patterns of immunoreactivity to T. solium antigens.** Fig 2A and 2D show calcified granulomas on Alizarin–Red stain evidencing extensive amorphous mineralization within fibrocalcific nodules and surrounded by a thin rim of reactive glial tissue. Fig 2B and 2C show extensive and intense immunoreactive areas to residual antigens with mAb TsW8 and TsV4, primarily localized within the calcified granuloma and displaying a dense staining pattern consistent with persistent antigenic deposits embedded in the fibrocalcific matrix. Fig 2E and 2F showed reactivity to T. solium antigens with mAbs TsV3 and TsE1 present into the perilesional brain tissue surrounding calcified granulomas. Scale black bar: 400 μm.

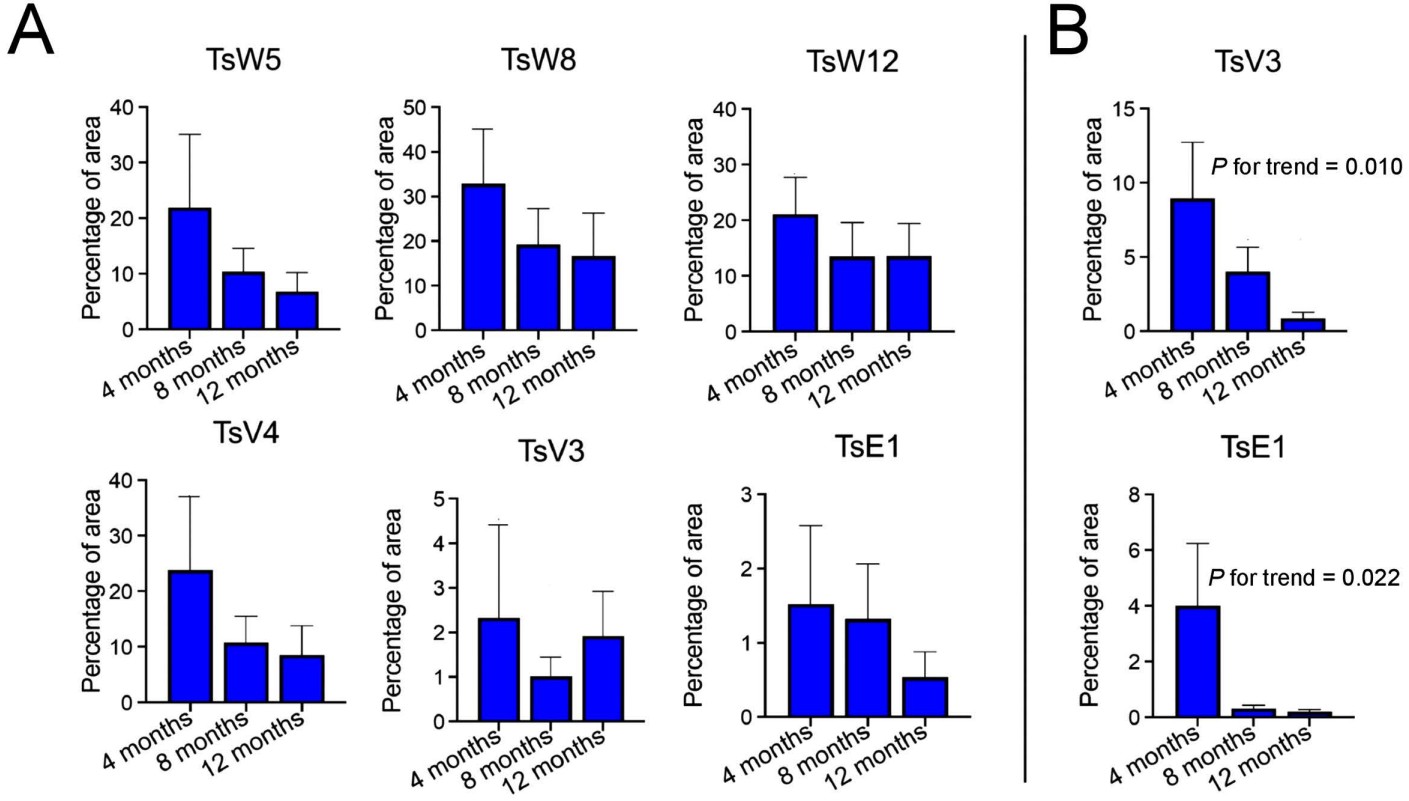

**Fig 3. Immunoreactive areas to *T. solium* antigens in calcified granulomas of naturally infected NCC pigs across time points after APT. (A)** Immunoreactive areas relative to the total area of the calcified granuloma; **(B)** Immunoreactive areas relative to the brain tissue area surrounding calcified granulomas (Bars: mean values; whiskers: standard errors). *P* values were obtained from non-parametric Cuzick tests for trend.

## Discussion

Neurocysticercosis represents a significant cause of acquired epilepsy in most of the world, and some calcified lesions are recognized as potential foci of seizure activity [17,27,28]. Using monoclonal antibody–based IHC, we showed the presence of residual parasite antigens within calcifications and in the surrounding pericystic brain tissue [25]. The nature of the study, however, does not allow to determine if residual parasite antigens may contribute to perilesional episodic inflammation and seizure recurrence occurring in calcified NCC [11,20–22,29].

Detection of cyst antigens in 65–80% of calcified granulomas is consistent with previous reports of residual parasite material in human and porcine NCC [19,21], indicating that remnants are frequently sequestered within calcifications. The novelty of this study lies in the use of our developed and standardized anti-*T. solium* mAbs in IHC assays for antigen detection and their distribution patterns [25]. Two distinct patterns emerged in calcified granulomas: antigens confined to calcified lesions (TsW5, TsW8, TsW12, and TsV4), and antigens present into adjacent parenchyma (TsV3 and TsE1). Both patterns are relevant to inflammatory response and epileptogenesis [17,27]. The variability in immunoreactivity observed across mAb-based IHC assays likely reflects underlying biological heterogeneity. Calcified NCC lesions represent a dynamic process in which parasite-derived antigens may persist differentially according to antigen stability, spatial localization, and epitope accessibility. Moreover, differences in monoclonal antibody targets may further contribute to variation in staining intensity and distribution.

Our findings lend support to the notion that parasite antigens entrapped within calcified granuloma lesions may continue to release immune responses through intermittent exposure, a process plausibly enabled by calcium resorption from the lesion, as previously proposed [30]. Furthermore, our longitudinal study in pigs indicates that the antigen burden within calcified granulomas and in the perilesional brain tissue (particularly with mAbs TsW8 and TsV3 respectively) declines over a 12-month period, suggesting a dynamic process [26]. The persistence of detectable parasite antigens may contribute to explain clinical reports of recurrent seizures and fluctuating perilesional edema in patients harboring calcified NCC months or years after lesion resolution [27]

In our study we only assessed the presence of *T. solium* antigens in the perilesional brain tissue, and we were unable to distinguish between episodes of intermittent antigen release and long–term antigen persistence in the perilesional brain tissue. [22]. These differences may be critical to understanding the neuropathology of NCC, since intermittent episodes of antigen leakage from calcified NCC lesions into the surrounding parenchyma can trigger transient induction of proinflammatory cytokines such TNF-a, IL6, and IL1-ß and the inflammatory response accompanied by perilesional edema and blood–brain–barrier disruption [31].On the other hand, persisting antigens that remain chronically in the surrounding brain tissue may represent long–term exposure and sustained microglia activation, leading to reactive gliosis, and synaptic reorganization that results in hyperexcitable cortical networks that serve as epileptogenic foci [31,32]. It could also explain damage distant from a brain cyst, as it seems to occur in the case of hippocampal temporal lobe atrophy and sclerosis in patients with NCC [33–35].

Image analysis showed that mAbs TsW5, TsW8, TsW12, and TSV4 exhibited similar recognition patterns in calcified granulomas, with consistently stronger reactivity than mAbs TsV3 and TsE1. These findings likely reflect the persistence of structural components during cyst degeneration, in line with histological reports of scolex remnants in calcified NCC [21,22]. In contrast, mAbs TsV3 and TsE1 detected residual antigens in surrounding brain parenchyma, suggesting that vesicular and excretory/secretory products may diffuse into the perilesional tissue, while structural antigens remain concentrated within calcifications. Overall, our findings align with those of Paredes et al. who described selected antigen recognition patterns of parasite components in viable brain cysts, such as the cyst wall and neck region with mAbs TsW5, TsW8, and TsW12, whereas mAbs TsV3, TsV4, ad TsE1 also recognize vesicular–fluid antigens [25]. Also, in the rat model of NCC, immunohistochemistry analysis showed IgG extravasation around cysts, evidence of antigen/BBB leakage in the perilesional tissue [36]. These observations provide valuable insights for the development of antigen-based IHC approaches aimed at improving the detection of residual cyst antigens within and surrounding calcified NCC lesions.

This study has drawbacks. The number of calcified granulomas analyzed was relatively small and limited by the availability of biopsies that met the eligibility criteria, which reduced the statistical power for comparisons of residual cyst antigen levels across post-treatment timepoints. Second, the use of a convenience sampling may limit the representativeness of specimens analyzed. Calcified lesions used in this study persisted for up to 12 months post-treatment; however, it remains uncertain whether antigens remnants can persist in older, long-standing calcifications, as observed in human NCC [21,22]. Third, our study was designed to optimize IHC and characterize antigen distribution in calcified granulomas; it did not include a predefined quantitative framework or untreated longitudinal groups with sufficient sample size to support statistical inference of antigen release over time. Future studies incorporating these elements are needed to directly test this mechanism. We also did not assess cross-reactivity with other microbial infections, as the study focused on characterizing residual antigen recognition using well-characterized anti–T. solium mAbs previously validated for species specificity and minimal cross-reactivity [25]. Finally, validation of IHC assays in human NCC is still required to confirm that the antigen recognition patterns observed in pigs mirror those in patients with epilepsy secondary to calcified NCC.

Altogether, this study demonstrates the presence and long-term persistence of residual parasite antigens in calcified NCC granulomas. Although antigens decline over time, they persist for at least 12 months post-treatment and can be detected by IHC with T. solium–specific monoclonal antibodies. Further investigations are warranted to elucidate the underlying mechanisms such as astrocytosis, microglial activation, and neuronal injury to clarify the role of residual cyst antigens

in disease progression. In addition, further studies should assess whether residual parasite antigens in calcified granulomas co-localize with markers of perilesional inflammation, glial activation and gliosis, and induce structural damage in neural cells or activate apoptotic pathways, in order to better elucidate the mechanisms underlying seizure development as has been suggested in NCC [37] and other neurological disorders [38,39]. Ultimately, our findings document features of antigen remnants in the pathological context of residual calcifications that may have impact on neuroinflammation, neurodegeneration, and seizure development, while also highlighting the potential use of our mAbs for NCC histological diagnosis of NCC and to differentiate from other neuroparasitic infections characterized by calcified or fibrotic sequelae.

## Supporting information

**S1 Table. Optimized conditions for IHC detection of *T. solium* cyst antigens, including dilutions of primary monoclonal antibodies, blocking agents, and secondary antibodies, validated on brain biopsies with viable brain cysts (positive controls) and uninfected tissue (negative controls for pericystic brain tissue).**
(DOCX)

**S2 Table. Immunoreactivity areas to *T. solium* cyst antigens (mean % of the total area±standard errors) as determined by mAb-based IHC assays in calcified granulomas from treated NCC pigs distributed across post-treatment time points.**
(DOCX)

**S1 Data. Study dataset containing immunorreactivity detection and immunoreactive areas using anti – *T. solium* antigens in calcified granulomas across post-treatment time points.**
(XLSX)

## Acknowledgments

We thank Dr. Sukwan Handali (Center for Diseases Control, Atlanta, Georgia, US) and Dr. Yesenia Castillo (Laboratory of Parasite Immunology, Department of Microbiology, School of Sciences, Universidad Peruana Cayetano Heredia, Lima, Peru) to provide the monoclonal antibodies used for the development and optimization of IHC assays for cyst antigen detection. Special thanks to Karen Arteaga (Cysticercosis Unit, Instituto Nacional de Ciencias Neurologicas, Lima, Peru) for their support during IHC assays.

Other members of the CWGP include Mirko Zimic, PhD, Armando E. Gonzalez, DVM, PhD; Seth E. O'Neal MD, MS, PhD (Coordination Board); Manuel Martinez, MD; Isidro Gonzalez, MD; Herbert Saavedra, MD; Sofia Sanchez, MD, MS (Instituto Nacional de Ciencias, Neurologicas, Lima, Peru); Saul Santivañez, MD, PhD; Holger Mayta, PhD; Yesenia Castillo, MS; Monica Pajuelo, PhD; Miguel A. Orrego, MS, PhD; Nancy Chile, PhD (Universidad Peruana Cayetano Heredia, Lima, Peru); Ana Vargas-Calla, DVM, MS; Eloy Gonzalez-Gustavson, DVM, MS, PhD; Luis A. Gomez-Puerta, DVM, MS; Cesar M. Gavidia, DVM, MPH, PhD; Teresa Lopez-Urbina, DVM, PhD (Universidad Nacional Mayor de San Marcos, Lima, Peru); Luz M. Moyano, MD, PhD; Ricardo Gamboa, MS; Percy Vilchez, MS; Claudio Muro (Cysticercosis Elimination Program, Tumbes, Peru); Sukwan Handali, MD, MS, PhD; John Noh (Center for Diseases and Control, Atlanta, Georgia, United States of America); John Friedland, PhD (St George, University of London, United Kingdom).

## Author contributions

**Conceptualization:** Luz M. Toribio, Gianfranco Arroyo, Hector H. Garcia, Javier A. Bustos.

**Data curation:** Lizziee B. Tello-Ccente, Gianfranco Arroyo.

**Formal analysis:** Gianfranco Arroyo.

**Funding acquisition:** Hector H. Garcia, Javier A. Bustos.

**Investigation:** Luz M. Toribio.

**Methodology:** Luz M. Toribio, Lizziee B. Tello-Ccente.

**Writing – original draft:** Luz M. Toribio, Gianfranco Arroyo.

**Writing – review & editing:** Manuela R. Verastegui, Robert H. Gilman, Theodore E. Nash, Hector H. Garcia, Javier A. Bustos.

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
