## [Decision Letter · Decision Letter 0]

3 Feb 2026

PNTD-D-25-02017

Monoclonal antibody-based immunohistochemistry reveals residual Taenia solium antigens in calcified granulomas from pigs with neurocysticercosis

Dear Dr. Arroyo,

Thank you for submitting your manuscript to PLOS Neglected Tropical Diseases. After careful consideration, we feel that it has merit but does not fully meet PLOS Neglected Tropical Diseases's publication criteria as it currently stands. Therefore, we invite you to submit a revised version of the manuscript that addresses the points raised during the review process.

Please submit your revised manuscript within by Apr 04 2026 11:59PM. If you will need more time than this to complete your revisions, please reply to this message or contact the journal office at plosntds@plos.org. Please include the following items when submitting your revised manuscript:

We look forward to receiving your revised manuscript.

Kind regards,

Marco Coral-Almeida, M.Sc., Ph.D.

Academic Editor

Francesca Tamarozzi

Section Editor

Shaden Kamhawi

co-Editor-in-Chief

Paul Brindley

co-Editor-in-Chief

**Journal Requirements:**

1) Please provide an Author Summary. This should appear in your manuscript between the Abstract (if applicable) and the Introduction, and should be 150-200 words long. The aim should be to make your findings accessible to a wide audience that includes both scientists and non-scientists. Sample summaries can be found on our website under Submission Guidelines:

- ® on page: 11.

- TM on page: 11.

4) We notice that your supplementary Tables are included in the manuscript file. Please remove them and upload them with the file type 'Supporting Information'. Please ensure that each Supporting Information file has a legend listed in the manuscript after the references list.

**Reviewers' Comments:**

Reviewer's Responses to Questions

**Key Review Criteria Required for Acceptance?**

**Methods**

-Are the objectives of the study clearly articulated with a clear testable hypothesis stated?

-Is the study design appropriate to address the stated objectives?

-Is the population clearly described and appropriate for the hypothesis being tested?

-Is the sample size sufficient to ensure adequate power to address the hypothesis being tested?

-Were correct statistical analysis used to support conclusions?

-Are there concerns about ethical or regulatory requirements being met?

Reviewer #1: This is a very well written manuscript, with a clearly explained objective, methodology and results section.

I only have (very) minor comments:

How were the 6 mAb selected from the 21 in house anti-Tsolium mAb? Why these 6?

Did the authors also check for inflammatation? Could they have kept sections from each block, eg the section closest to the section use for IHC?

Line 104 and 133-134: presence OF…

Reviewer #2: The methodology is well described

**Results**

-Does the analysis presented match the analysis plan?

-Are the results clearly and completely presented?

-Are the figures (Tables, Images) of sufficient quality for clarity?

Reviewer #1: This is a very well written manuscript, with a clearly explained objective, methodology and results section.

Reviewer #2: the results is well described

**Conclusions**

-Are the conclusions supported by the data presented?

-Are the limitations of analysis clearly described?

-Do the authors discuss how these data can be helpful to advance our understanding of the topic under study?

-Is public health relevance addressed?

Reviewer #1: The discussion is also well elaborated, could perhaps be shortened a bit, and includes a relevant limitation section.

Reviewer #2: The conclusion support the presented data

**Editorial and Data Presentation Modifications?**

Reviewer #1: (No Response)

Reviewer #2: -

**Summary and General Comments**

Reviewer #1: This manuscript decribes the analyses of calcified cysts collected from NCC pig brains (of treated pigs), using monoclonal antibodies to identify and localise the presence of specific antigens, at different time points. Results may contribute greatly to the understanding of inflammatory processes (and seizures) around calcified cysts in human.

This is a very well written manuscript, with a clearly explained objective, methodology and results section. The discussion is also well elaborated, could perhaps be shortened a bit, and includes a relevant limitation section.

I only have (very) minor comments:

How were the 6 mAb selected from the 21 in house anti-Tsolium mAb? Why these 6?

Did the authors also check for inflammatation? Could they have kept sections from each block, eg the section closest to the section use for IHC?

Line 104 and 133-134: presence OF…

Reviewer #2: The article is well written and discussed, the methodology is acceptable, and both the methodology and results are well described. The study findings may help in understanding the neuropathology of NCC.

Some comments need to be addressed by the authors.

-Why did the authors not test for cross-reactivity using positive samples representing microbial infections?

-How does the author explain the presence of immunoreactive variabilities among the used antigens?

-The author needs to compare (with statistical significance) lesional and peri-lesional concentrations of the antigens at different time points in treated and non-treated NCC pigs to support the hypothesis of release of cyst antigens.

- Is this the first evidence of cyst antigen presence and persistence in calcified lesions and surrounding cerebral tissues? If not, state other similar evidence and comment.

-The authors claim that their findings support the hypothesis that residual parasite antigens persisting in calcified lesions and intermittently released in the perilesional brain tissue may elicit episodic inflammatory responses. However, they did not test for these hypotheses.

PLOS authors have the option to publish the peer review history of their article (what does this mean?). If published, this will include your full peer review and any attached files.

Reviewer #1: No

Reviewer #2: No

**Figure resubmission:** While revising your submission, we strongly recommend that you use PLOS’s NAAS tool (https://ngplosjournals.pagemajik.ai/artanalysis) to test your figure files. NAAS can convert your figure files to the TIFF file type and meet basic requirements (such as print size, resolution), or provide you with a report on issues that do not meet our requirements and that NAAS cannot fix.
---

## [Decision Letter · Decision Letter 1]

4 May 2026

Dear Arroyo,

We are pleased to inform you that your manuscript 'Monoclonal antibody-based immunohistochemistry reveals residual Taenia solium antigens in calcified granulomas from pigs with neurocysticercosis' has been provisionally accepted for publication in PLOS Neglected Tropical Diseases.

Best regards,

Marco Coral-Almeida, M.Sc., Ph.D.

Academic Editor

Francesca Tamarozzi

Section Editor

Shaden Kamhawi

co-Editor-in-Chief

Paul Brindley

co-Editor-in-Chief

Reviewer's Responses to Questions

**Key Review Criteria Required for Acceptance?**

**Methods**

-Are the objectives of the study clearly articulated with a clear testable hypothesis stated?

-Is the study design appropriate to address the stated objectives?

-Is the population clearly described and appropriate for the hypothesis being tested?

-Is the sample size sufficient to ensure adequate power to address the hypothesis being tested?

-Were correct statistical analysis used to support conclusions?

-Are there concerns about ethical or regulatory requirements being met?

Reviewer #1: the authors have sufficiently replied to the reviewers comments, I have no further comments.

Reviewer #2: (No Response)

**Results**

-Does the analysis presented match the analysis plan?

-Are the results clearly and completely presented?

-Are the figures (Tables, Images) of sufficient quality for clarity?

Reviewer #1: the authors have sufficiently replied to the reviewers comments, I have no further comments.

Reviewer #2: (No Response)

**Conclusions**

-Are the conclusions supported by the data presented?

-Are the limitations of analysis clearly described?

-Do the authors discuss how these data can be helpful to advance our understanding of the topic under study?

-Is public health relevance addressed?

Reviewer #1: the authors have sufficiently replied to the reviewers comments, I have no further comments.

Reviewer #2: (No Response)

**Editorial and Data Presentation Modifications?**

Reviewer #1: (No Response)

Reviewer #2: (No Response)

**Summary and General Comments**

Reviewer #1: the authors have sufficiently replied to the reviewers comments, I have no further comments.

Reviewer #2: The authors addressed the reviewers' comments

PLOS authors have the option to publish the peer review history of their article (what does this mean?). If published, this will include your full peer review and any attached files.

Reviewer #1: No

Reviewer #2: No

---

## [Editor Report · Acceptance letter]

Dear Arroyo,

We are delighted to inform you that your manuscript, "Monoclonal antibody-based immunohistochemistry reveals residual Taenia solium antigens in calcified granulomas from pigs with neurocysticercosis," has been formally accepted for publication in PLOS Neglected Tropical Diseases.

Best regards,

Shaden Kamhawi

co-Editor-in-Chief

Paul Brindley

co-Editor-in-Chief
